# Evaluation of Lake Fragmentation and Its Effect on Wintering Waterbirds in Poyang Lake, China

Muhammad Suliman [1,†], Wenyou Deng [1,†], Qingming Wu [1,*], Tariq Ahmad [1], Xueying Sun [1], Debela Megersa Tsegaye [1], Muhammad Sadiq Khan [2], Ngo Thi Kieu Trang [1] and Hongfei Zou [1,*]

1    College of Wildlife and Protected Area, Northeast Forestry University, No. 26, Hexing Road, Harbin 150040, China; suliman0370@gmail.com (M.S.); dengwenyou@nefu.edu.cn (W.D.); tariq.zoologist@yahoo.com (T.A.); sxytuvwz@126.com (X.S.); megtsegaye@gmail.com (D.M.T.); ngothikieut-rang96@126.com (N.T.K.T.)
2    Key Laboratory of Vegetation Restoration and Management of Degraded Ecosystems, South China Botanical Garden, Chinese Academy of Sciences, Guangzhou 510650, China; khan_eco@scbg.ac.cn
*    Correspondence: qingmingwu@nefu.edu.cn (Q.W.); hongfeizou@163.com (H.Z.); Tel.: +86-15945685507 (Q.W.); +86-13503636269 (H.Z.)
†    These authors contributed equally to this research work.

**Abstract:** This study was designed to determine the fragmentation of sub-lakes in winter and its effects on wintering waterbirds in Poyang Lake. Poyang Lake becomes fragmented in winter, which forms many seasonal sub-lakes every year, and have different environmental characteristics. These sub-lakes significantly impact winter bird habitats and result in susceptibility to various changes, because birds have different distribution responses. A total of 24 sub-lakes were surveyed from one to five vantage points using point count methods in each sub-lake with binoculars, monocular, and a spotting scope for four consecutive winter seasons. The multi-site dissimilarity Sorensen index measures overlapped between two populations, and the R software "iNEXT" package was used to evaluate the sample coverage test of the study area. We observed 58 wintering waterbird species belonging to 9 orders and 15 families from 2016 to 2020. Spearman correlation analysis showed that the species richness of wintering waterbirds was significantly positively correlated with the sub-lake areas and associated with the richness of habitat type. The WNODF analyses were considerably correlated for sites of waterbirds, mainly with the abundance of forage and conservation of habitat form. The outcomes of this study showed that Maying Lake has the highest local beta diversity, whereas Dacha Lake has the lowest local beta diversity contribution (0.007). This study's findings demonstrate Poyang Lake's role in waterbird the habitat suitability of waterbirds, especially for foraging and conservation.

**Keywords:** lake fragmentation; wintering waterbirds; Poyang Lake; China



## 1. Introduction

As a natural reserve of global significance, wetland ecosystems have been broadly recognized as playing a key role in supporting global biodiversity by providing vital habitat for frequent wildlife species [1]. Waterbirds, a group of bird species living together mostly dependent on water, are widely distributed worldwide. Waterbirds entirely or partially depend on wetlands for various activities such as foraging, loafing, and molting [2]. However, their populations are being influenced worldwide and have begun to fluctuate due to widespread human activities and environ mental change [3]. The economic development and the improvement of people's living standards have changed the original land use mode and environment characteristics, and the continuous natural landscape has been divided into different green patches, aggravating the impact of habitat fragmentation, alien species invasion, and human interference, resulting in an island effect of biodiversity [3]. Birds are an important component of biodiversity and an indicator species for environmental

monitoring [4]. They are common animal groups found everywhere and are important in ecosystem service functions [5]. Previous studies have shown that the fragmentation of natural landscape will lead to the homogenization of birds' functional diversity and the decline in their role in ecosystem service functions, as waterbirds' reproductive life history is linked to home range, which is particularly noteworthy [6]. Beta diversity refers to the difference in species composition between different communities, which is widely used in conservation biology. For a given level of regional species richness, as beta diversity increases, individual localities differ more markedly from one another and a smaller proportion of the species occur in the region [7]. Spatial turnover may be ascribed to species replacement between assemblages fueled by environmental straining and dispersal ability, allowing species to track their appropriate environments, while nestedness results from species loss or gain associated with ordered extinction–colonization dynamics [8]. Determining and decomposing β-diversity results in testing hypotheses related to biodiversity distributions and assembly processes of biotic groups [8,9], providing visions for the organization, preservation, and restoration of biodiversity [10,11]. It has subsequently been frequently emphasized [8,12]. However, the effect of nestedness on similarities among biotas has been known for a long time [13]. Whittaker's definition emphasizes the variation in species composition among sites within a geographic area. Baselga decomposes beta diversity into the component of species turnover and the nested component [14]. Species turnover indicates the replacement of species among different communities (i.e., changes in species composition among local communities). A nested component demonstrates that when the species richness is arranged in order along a gradient, the community with fewer species is a subset of the community with more species [2], or it can be explained as the loss of species at specific sites where there is a subset of sites with more prosperous species. Beta diversity is affected by deterministic and random factors, such as environmental filtering, geographical distance, intraspecific competition, and human interference [15]. Theoretically, this is important in conservation biology and is often applied to biogeographic zoning and reserve site selection. If nested components are dominant, the regions with high species richness have high protection levels. Otherwise, it means that the contribution of all studied areas to beta diversity is similar [16]. If the spatial turnover constituent is the main pattern of beta diversity, more protected parts are essential to conserve regional biodiversity. In contrast, if the nestedness component is the main pattern, a large protected area comprising a high species richness could be sufficient [13]. Thus, beta diversity is an essential tool for conservation planning.

In spatial turnover, species are replaced from one site to another due to dispersal or niche processes, which may occur contemporaneously or in the past. Contrary to turnover, nestedness-resultant results ($\beta_{nes}$) are determined by species losses and gains in nested subsets caused by contemporary or historical processes, including passive sampling, selective extinction, selective colonization, and habitat nestedness [17,18]. Range shifts and phenological modification are two practices by which organisms respond to environmental warming. Understanding the mechanisms that drive these changes is the key to the best conservation and administration of population supervision, locally and across the species range [19].

To explain the nested pattern, the passive sampling hypothesis including birds extinction hypothesis, selective colonization hypothesis, and habitat nestedness hypothesis, were explored by many scholars. Thus, this hypothesis can reveal the ecological process [20]. Passive sampling (i.e., sampling from the species pool in proportion to species abundances) describes a situation in which rare species are diminished in the community compared to abundant species within a particular area [21,22]. Previous studies hypothesized and give predictions that when systems are experiencing species loss or fauna relaxation (downsizing), the site should be the primary driver of nestedness because species with large minimum area requirements have greater extinction risk, and a predictable sequence of extinction might occur concerning island size [23,24]. Also, differential extinction could result in nestedness if species exhibit specific area requirements and island areas differ

in the analysis system. In such cases where local extinction affects community structure, species tend to become extinct due to their specific extinction risk or vulnerabilities. For the selective colonization hypothesis, species differ in their ability to colonize distant sites [25]. Differential colonization could result in nestedness if highly vagile species occupy most islands and less vagile species inhabit only the closer, larger islands [25]. Finally, the habitat nestedness hypothesis considers the nestedness of species assemblages and their reliance on the distribution of nested habitats [26].

China's largest freshwater lake, Poyang Lake is located on the south bank of the middle Yangtze River. Although Poyang Lake supports the conservation of natural ecosystems and multi-species habitats, land use, climate, and environmental characteristics have affected bird species' survival and spatial distribution [27].

Due to the natural hydrological law, Poyang Lake forms many sub-lakes with different areas in the winter period every year [28]. Different sub-lakes equivalent and fragmented to an island forms, have different environmental characteristics that provide habitat for wintering waterbirds, and birds have different distribution responses based on forage and bird preferences. Even though many previous studies have focused on the species composition and spatial distribution pattern of the avian communities in Poyang lake while the sub-lakes act as lake island due to fragmentation in the winter season, waterbird communities have received less attention. Thus, this study discusses the lake island effect due to fragmentation of lakes in the winter season and its effects on the wintering waterbird community structure by beta diversity partitioning in Poyang Lake. In addition, this study assesses the relationship between bird species richness and the sub-lake areas, between the sub-lake area and the habitat type's richness.

## 2. Material and Methods

### 2.1. Study Area and Study Design

The current study areas were sub-lakes located in two national nature reserves, Poyang Lake National Nature Reserve (PNNR) and Nanji National Nature Reserve (NNNR), and one provincial nature reserve, Duchang Provincial Nature Reserve (DPNR) in the Poyang Lake Basin (Figure 1). The wintering waterbird data were from four winter seasons: 17–31 January in 2016, 9–26 January in 2017, 19–31 January in 2018, and 9–26 January in 2019, when the population of birds was relatively stable [27,28]. This study on sub-lakes with an area greater than 2 km$^2$ was selected based on geographical representation for verifying the waterbird community turnover component and nested component. Accordingly, 24 sub-lakes were surveyed from one to five vantage points using point count methods in each sub-lake with binoculars, monocular, and a spotting scope for four consecutive winter seasons, commonly known as Bibby's bird census counting methods [29]. The R software 4.2.3. "iNEXT" package is used to evaluate the sample coverage test of the study area [24].

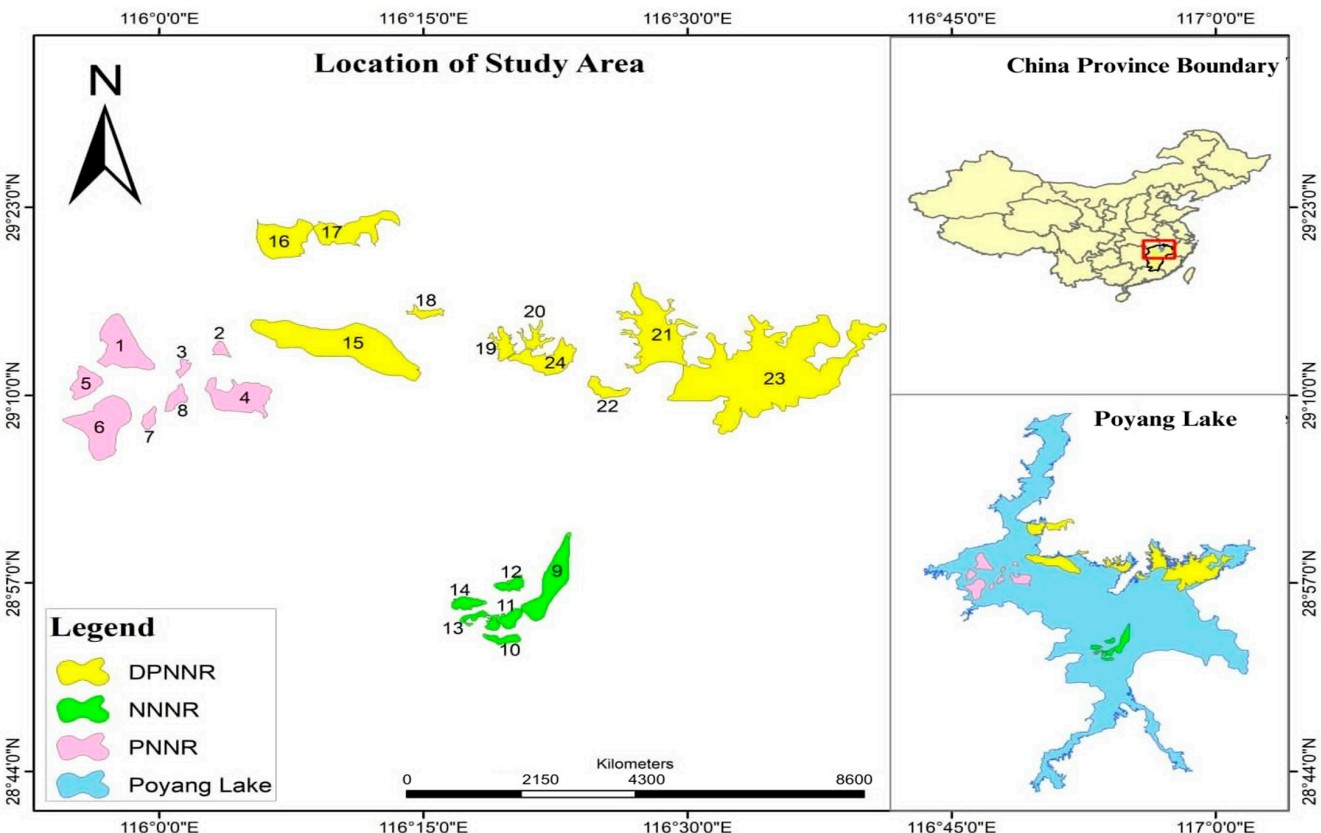

**Figure 1.** Location of study area: Poyang lake, Jiangxai Province, China. Note: Bang hu—1, Meixi Hu—2, Zengmizhou Hu—3, Dacha Hu—4, Sha Hu—5, Dahuchi—6, Chang Hu—7, Zhonghuchi Hu—8, Dong Hu—9, Fengwei Hu—10, Baisha Hu—11, Beishen Hu—12, Sani Hu—13, Zhanbei Hu—14, Jishan—15, Maying Hu—16, Xinmiao Hu—17, DaMian Hu—18, Nanxi Hu—19, Zhudeye Hu—20, Shipai Hu—21, ZhuTong Hu—22, Poyang Hu—23, HuaMiao Hu—24. DPNNR: Duchang Provincial National Nature Reserve, NNNR: Nanjishan Wetland National Nature, PNNR: Poyang Lake National Nature Reserve. The red box is the area where the Poyang Lake is located in the Main land of China.

### 2.2. Methods and Data Analysis

The species composition of wintering waterbirds in 24 sub-lakes was compared by beta diversity to understand the difference of species composition on wintering waterbirds in detail. To calculate the multi-site dissimilarity Sorensen index, including the total beta diversity ($\beta_{sor}$), spatial turnover ($\beta_{sim}$), and nesting component ($\beta_{nes}$), the beta part in R was used [24]. Following Baselga, the total beta diversity was broken down into two separate components: the species turnover component, which is measured by the Simpson dissimilarity index ($\beta_{sim}$), and the nestedness component ($\beta_{nes}$) [30].

Thus, $\beta_{sor}$ was decomposed into $\beta_{sim}$ (spatial turnover) and $\beta_{nes}$ (nestedness), $\beta_{sor} = \beta_{sim} + \beta_{nes}$.

$$\beta_{nes} = \frac{\max(b,c) - \min(b,c)}{2a + b + c} \frac{a}{a + \min(b,c)}$$

$$\beta_{nes} = \frac{\min(b,c)}{a + \min(b,c)}$$

where *a* is the number of shared bird species among two areas, and *b* and *c* are the number of species only present in the first and second areas. The Sorensen index ranges from 0 (no species are shared between the two areas) to 1 (all species are common between the two areas) [31]. The ratio of $\beta_{sim}/\beta_{sor}$ can evaluate the relative contribution of turnover to composition dissimilarity in each group ($\beta_{ratio} = \beta_{sim}/\beta_{sor}$) when $\beta_{ratio} < 0.5$, the beta

diversity is mainly determined by the nestedness component ($\beta_{nes}$) or species replacement between assemblages (i.e., $\beta_{sim}$ or turnover component). Conversely, when $\beta_{ratio} > 0.5$, the beta diversity is mainly determined by the spatial turnover component ($\beta_{sim}$) or the nestedness component ($\beta_{nes}$) [30,32–34].

*2.3. Statistical Analysis*

Spearman correlation analysis was used to analyze the relationship between the bird species richness and the sub-lake areas and between the sub-lake area and the habitat birds' richness [35]. Habitat variables such as species diversity and richness were measured by sub-lake area, distance from the nearest large sub-lake, and the number of waterbird habitat types. The distance between sub-lakes reflects the straight-line distance of birds across nearby space [18]. As suggested in the previous study [26], the habitat types were divided into 4 types, namely mudflats, vegetation, waterbody, and sands (bare land). Spearman correlation analysis was used to analyze the factors affecting the nesting pattern of waterbirds in Poyang Lake [16,26,36,37]. The WNODF analysis method can analyze not only the whole matrix, species, and research site the whole matrix, but also analyze species and research sites. It is insensitive to the filling size of the matrix as it can avoid type I errors, unlike NODF, and can distinguish species incidence from species composition [17]. Thus, we used the metric WNODF to estimate nestedness in this study.

## 3. Results

*3.1. Analysis of Winter Water Birds Composition and Species Richness*

Fifty-eight wintering waterbird species belonging to 9 orders and 15 families were observed at the study sites during the five winter seasons from 2016 to 2020. Overall sample coverage was $0.96 \pm 0.02$, indicating enough sample size. Spearman correlation analysis showed significant positive relationships between the species richness of wintering waterbirds and the sub-lake areas ($r = 67.55$, $p < 0.01$), and between the sub lake area and the habitat bird's richness ($r = 64.25$, $p < 0.01$) (Table 1).

**Table 1.** Results of nesting analysis using NODF program for species by site matrix of wintering waterbirds in Poyang Lake, China.

| Metrics | $N_{obs}$ | $N_{exp}$ (SD) | Z-Value | $p$ |
|---------|-----------|----------------|---------|-----|
| WNODF | 33.19 | 64.63 (2.68) | −11.75 | 0.001 |
| WNODFc | 34.68 | 74.34 (3.09) | −12.85 | 0.001 |
| WNODFr | 32.88 | 62.64 (3.09) | −9.64 | 0.001 |

Abbreviations: NODF is used to measure the nesting pattern of birds; Nobs is observed NODF for sites. $N_{exp}$ (SD) is the expected NODF for sites with a standard deviation. $p$ is Monte Carlo-derived probabilities. WNODFc is a weighted nestedness metric based on overlapping and decreasing fill, columns, or sites. WNODFr is a weighted nestedness metric based on overlapping and decreasing fill, rows, or species.

*3.2. Analysis of Beta Diversity of Poyang and Sub-Lake Waterbirds*

Beta diversity analysis showed that the species diversity of wintering waterbirds in sub-lakes of Poyang Lake ($\beta_{sim} = 0.464$) were greater than nested ones ($\beta_{nes} = 0.364$). The bird species composition in each sub-lake was different in the nestedness analysis using WNODF, which showed that the waterbird numbers ($N_{obs} = 33.19$) were significantly lower than expected from the null model ($N_{exp} = 64.63$, Z-value = −11.75, $p = 0.001$) as given in Table 1. This indicated that there were more waterbird communities in non-nested sites, which also means that the observed communities were less in number in the nested sites than expected by null matrices (based on proportional-row and proportional-column constraints with 1000 randomizations).

Spearman correlation analysis (Table 2) showed that in the lake area, the habitat types' richness was significantly correlated with nesting rank ($r = −0.644$, $p < 0.01$ and $r = −0.646$, $p < 0.01$), while the distance from the nearest large sub-lake was not significantly correlated with nesting rank ($r = 0.495$, $p > 0.05$). Thus, the factors affecting the nesting pattern of

waterbirds in the study area are related to sub-lake area and habitat bird richness, and have no correlation with sub-lake connectivity. This conforms to the selective extinction hypothesis and habitat nestedness hypothesis, and does not support the particular colonization hypothesis. The comprehensive results showed that small- and medium-sized sub-lakes are also experiencing biodiversity loss.

**Table 2.** Characteristic parameters of sub-lakes in Poyang Lake.

| Sub-Lake Site | Species Richness | Sample Coverage | Sub Lake Area (km²) | Sub Lake Connectivity (m) | Habitat Type Richness | Nested Rank |
|---|---|---|---|---|---|---|
| Zhanbei | 30 | 0.99 | 140 | 716.00 | 4 | 1 |
| Bang | 28 | 0.98 | 260 | 800.00 | 4 | 2 |
| Baisha | 26 | 0.99 | 41 | 138.00 | 4 | 3 |
| Dacha | 24 | 0.99 | 180 | 1092.25 | 4 | 4 |
| Sanniwan | 23 | 1.00 | 16 | 445.00 | 4 | 5 |
| Sha | 23 | 1.00 | 7 | 1332.65 | 3 | 5 |
| Poyang | 23 | 1.00 | 129 | 664.00 | 4 | 5 |
| Fengwei | 20 | 0.96 | 13 | 280.00 | 3 | 6 |
| Zhudeye | 20 | 1.00 | 5 | 177.00 | 3 | 6 |
| Chang | 19 | 0.97 | 20 | 1165.00 | 4 | 7 |
| Zhonghuchi | 19 | 0.98 | 4 | 1044.56 | 4 | 7 |
| Shipa | 19 | 1.00 | 34 | 412.00 | 4 | 7 |
| Beishen | 18 | 0.99 | 3 | 228.00 | 4 | 8 |
| Huamiao | 18 | 0.99 | 13 | 334.00 | 4 | 8 |
| Zengmizhou | 16 | 0.98 | 2 | 1812.00 | 3 | 9 |
| Maying | 16 | 0.97 | 18 | 1331.26 | 3 | 9 |
| Jishan | 16 | 0.99 | 53 | 4071.79 | 3 | 9 |
| Xinmiao | 15 | 1.00 | 17 | 1331.26 | 3 | 10 |
| Damian | 15 | 1.00 | 3 | 4071.79 | 3 | 10 |
| Meixi | 14 | 0.99 | 2 | 1129.34 | 3 | 11 |
| Nanxi | 14 | 1.00 | 5 | 219.00 | 3 | 11 |
| Dong | 13 | 0.99 | 18 | 224.00 | 3 | 12 |
| Dahuchi | 13 | 1.00 | 2 | 1650.00 | 3 | 12 |
| Zhutong | 13 | 1.00 | 5 | 2255.00 | 3 | 12 |

This study showed that Maying Lake has the highest local beta diversity contribution (0.062), whereas Dacha Lake has the lowest local beta diversity contribution (0.007) (Table 3).

**Table 3.** Local beta diversity contribution of 24 sub-lakes of Poyang Lake.

| Site | Local Beta Diversity Contribution | Site | Local Beta Diversity Contribution |
|---|---|---|---|
| Maying | 0.062 | Zhudeye | 0.043 |
| Sanniwan | 0.061 | Changhuchi | 0.039 |
| Damian | 0.059 | Zengmizhou | 0.038 |
| Bang | 0.059 | Meixi | 0.036 |
| Dong | 0.058 | Beishen | 0.033 |
| Huamiao | 0.052 | Zhanbei | 0.028 |
| Xinmiao | 0.051 | Baisha | 0.025 |
| Sha | 0.049 | Poyang | 0.023 |
| Nanxi | 0.048 | Jishan | 0.021 |
| Zhutong | 0.046 | Shipa | 0.020 |
| Fengwei | 0.045 | Dahuchi | 0.010 |
| Zhonghuchi | 0.044 | Dacha | 0.007 |

## 4. Discussion

This study demonstrates that the area of each sub-lake has a significant positive correlation with the richness of waterbird species, indicating that the larger sub-lake area

showed higher bird richness as it offers more suitable habitat, which is consistent with the theory of island biogeography [21,38]. The study highlighted the importance of individual sub-lakes through the Local Contribution to Beta Diversity (LCBD) metric [39]. According to the beta diversity analysis, the turnover processes significantly influenced the overall beta diversity of wintering waterbirds of the Poyang lake and the sub-lakes. Some sub-lakes demonstrated higher ecological uniqueness in contributing to the beta diversity, while others showed lower contributions. The turnover dominance showed variations in bird species among different sub-lakes. Beta diversity analysis showed that the diversity turnover components of waterbirds in Poyang Lake ($\beta_{sim}$ = 0.464) were more significant than nested components ($\beta_{sne}$ = 0.364), and the bird species composition in each sub-lake was different. Interestingly, even smaller sub-lakes play a role in protecting waterbirds, which highlights the importance of diverse habitats within an ecosystem.

This study showed that Maying Lake has the highest local beta diversity contribution (0.062), whereas Dachahu Lake has the lowest local beta diversity contribution (0.007). In line with the habitat nestedness hypothesis, the factors affecting the nesting pattern of waterbird diversity in Poyang Lake are particularly associated with the habitat type's richness. The nestedness analysis showed a significant nesting pattern distribution of wintering waterbirds in Poyang Lake. Similarly, the study sites (WNODFc, $p < 0.01$) also produced a significant nesting pattern, which further supports the habitat nesting hypothesis, which is consistent with the previous studies [18,21]. Therefore, habitat nesting, recognized as a concise explanation for species nesting distributions, focuses directly on the relationship between species and their habitats without delving into population dynamics or life histories and directly points to the relationship between species and their habitats [40,41]. While this study did not prove the dominance of environmental factors, such as area or habitat diversity, in determining nesting patterns, it strongly supported the idea that larger study areas and landscapes with high heterogeneity could contribute significantly to protecting and maintaining waterbird diversity within the Poyang Lake ecosystem. It underscored the significance of diverse habitats and landscape characteristics in species conservation of wintering waterbird populations in wetland ecosystems.

## 5. Suggestions

The results of the study showed that many waterbirds spend the winter at Poyang Lake. Thus, a comprehensive conservation action involving harmonized and protected potential wintering sites should be implemented. Many wintering waterbirds should be more monitored and need more attention. These lakes were the winter sites that harbored the most species of birds, including endangered birds. It is necessary to enlighten the public about the importance of bird conservation.

Similarly, to better understand the variation of bird distribution and impact of landscape patterns in Poyang Lake, future research should pay more attention to using remote sensing technology and landscape metrics. Wetland restoration projects should emphasize on the northern edge, easterly on the northerly edge, and eastern aspect on the northern and eastern direction. The Poyang Lake Basin's northern edge and eastern areas have more blank habitats and potential suitable habitats and can carry out more wetland restoration projects for wintering waterbirds.

## 6. Conclusions

This study investigated the diversity of waterbirds, evaluated the turnover component in the nested component of waterbird communities and nesting patterns in the diversity of wintering waterbirds in Poyang Lake and wintering sub-lakes, and investigated the waterbird's beta and functional diversity. A total of 58 wintering waterbird species were grouped into nine orders and 15 families and were recorded in the study sites during the five winter seasons from 2016 to 2020.

The lake island effect significantly influences bird species richness, correlating strongly with the sub-lake area, which, in turn, substantially correlates with habitat type richness.

Investigating waterbird nesting patterns indicates a significant correlation between sub-lake area, richness of habitat types, and nesting rank, aligning with selective extinction and habitat nestedness hypotheses. Beta diversity analysis underscores Maying Lake's substantial contribution to local beta diversity. Overall, the study provides valuable insights into the intricate dynamics of wintering waterbird communities in the Poyang Lake Basin, elucidating spatial distributions, habitat preferences, and ecological mechanisms influencing their presence and nesting patterns.

**Author Contributions:** Conceptualization, M.S., W.D., H.Z. and Q.W.; methodology, M.S., T.A., D.M.T. and W.D.; software. M.S., T.A. and D.M.T.; validation, Q.W. and X.S.; formal analysis, X.S. and M.S.K.; investigation, M.S., M.S.K., T.A., X.S., N.T.K.T. and Q.W.; resources M.S., M.S.K. and T.A.; data curation, M.S., H.Z., Q.W., T.A. and M.S.K.; writing—original draft, M.S.; writing—review and editing, Q.W., M.S., W.D., D.M.T., T.A. and M.S.K.; visualization, M.S., W.D., T.A. and D.M.T.; supervision, Q.W. and H.Z.; project administration, Q.W. and H.Z.; funding acquisition, Q.W. and H.Z. All authors have read and agreed to the published version of the manuscript.

**Funding:** The current study was supported by the National natural science foundation of China (grant number, 32271557), the national key research and development program of China (grant number, 2023YFF1305000), the Third comprehensive Scientific Investigation project in Xinjiang (grant number, 2021xjkk1200) and the fundamental research funds for the central Universities (grant number, 2572022AW19).

**Institutional Review Board Statement:** Not applicable.

**Informed Consent Statement:** Not applicable.

**Data Availability Statement:** Data are contained within the article.

**Acknowledgments:** We are very thankful to National Natural Science Foundation of China, The national key research and development program of China and the Third comprehensive Scientific Investigation project in Xinjiang. and the fundamental research funds for the central Universities, for funding and supporting. We are also very grateful for the comments and suggestions made by the anonymous reviewers.

**Conflicts of Interest:** The authors declare no conflicts of interest.

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
