# Peer review of "Evaluation of Lake Fragmentation and Its Effect on Wintering Waterbirds in Poyang Lake, China"

_diversity, doi:10.3390/d16030154_

Round 1

Reviewer 1 Report

Comments and Suggestions for Authors

Comments

-        16-18 (abstract): is there an official definition on sub-lakes ? Which are the different environmental characteristics ?

-        25-28 (abstract): I think is better to avoid in abstract the detailed numbers resulting by statistics (e.g. Nexp 26 = 64.63, Z-value = -11.75, p = 0.001) -        29-30 (abstract): maybe the phrase is better: especially regarding foraging and from a conservation point of view -        109-111: please give a citation of this affirmation -        112-115: see 16-18 -        115-118: please give citations of this affirmation -        119-121: it is unclear what is the relationship between bird species richness and the sub-lake areas. I think you should be more specific. -        120-121: what is the habitat types richness ? -        135-136: please give a citation for this software -        137 (figure 1): what are the following abbreviations: DPNNR, NNNR, PNNR,…. ? -        167: which habitat variables did you focused and how did you measured these variables ? -        171-175: why do you analyze the nesting pattern ? Your study focuses on wintering birds… -        191-195: The phrase is not so clear. Please rephrase the sentence. -        218: table 2: I think characteristic parameters of sub-lakes should not limited to area, connectivity and habitat type richness. Main habitat types should be important regarding their areas and accessibility for birds foraging, resting etc. It seems that the analysis is quite simplistic and incomplete. In this case, maybe a remote sensing study of all components / variables should be done. This may be even more necessary to capture the variations of the accessible foraging ares, respectively of the habitats, influenced by water level fluctuations. You didn't write anything about the water level in the lake -        221-226: this part is in fact a conclusion or should be moved to the abstract -        226-229: this conclusion is obvious -        220 (discussion): As before, nesting was discussed in a study that referred to wintering… -        275-277: my suggestion regarding using of remote sensing technology is also yours -        292-299 (conclusions): there are results that are not presents in the study ! -        Conclusions: there are repeated some information from results and discussion -        311-334:  there are results that are not presents in the study ! There are also some useless data regarding species geographic origin.. -        Only on the used variables (sub-lake areas, distances, number of the main habitat types..) this study could not elucidates the intricate dynamics of wintering waterbird communities 336 in the Poyang Lake Basin  

Suggestions

1.      Redoing the entire analysis, based on other variables of the feeding and resting habitat, but also on possible anthropogenic disturbance factors, water level fluctuation during winter etc.  

Using a complete analyze based on remote sensing or GIS models 

Comments on the Quality of English Language

Moderate editing of English language required.

Author Response

Thank you Reviewer i have done point by response

Reviewer 2 Report

Comments and Suggestions for Authors

This is a study of the community ecology of waterbirds at Poyang Lake, China, based on repeated surveys of birds at 24 sub-lakes over five consecutive winters. The study documented 58 different species of waterbirds across the sub-lakes.

Unfortunately, the issues with the language make the manuscript difficult to understand. I struggled to follow the aims and reported results of the study. The text would be more informative if key statements were supported by citations of relevant literature. I have given examples of this in my specific comments (see below).

Specific comments:

Line 10: It is not clear what the "lake-island effect" means in this context. You should add a brief definition so that the reader can better understand the context and aims of your study.

Line 12: To improve the language, "Total" should be amended to "A total of".

Line 14: To make the abstract more informative for the reader, you should state what you used the multi-site dissimilarity Sorensen index and R software "iNEXT" package for. What questions were you trying to answer?

Line 16: "Spearman Correlation was significantly positively for species richness." This result is unclear. What did species richness show a positive correlation with?

Line 16: What is "WNODF"? You need to define any acronym on its first use, otherwise how will the reader know what it means?

Line 20: To improve the language, the first letter of the new sentence should be capitalized.

Lines 26-27: The reference by Soga & Koike (2012) is not appropriate as this paper refers to butterflies, whereas your sentence makes a claim related to birds. You should replace the original reference with a more appropriate reference that supports your statement, for example: Montaño‐Centellas, F.A., Loiselle, B.A. & Tingley, M.W. (2021). Ecological drivers of avian community assembly along a tropical elevation gradient. Ecography, 44, 574-588.

Lines 27-28: Here, you should include a reference that defines waterbirds, so that readers understand the scope of your study. For example, this paper defines waterbirds as those avian species that live on or around aquatic habitats: Boere, G.C., Galbraith, C.A., & Stroud, D.A. (2007). Waterbirds around the world. Edinburgh, TSO Scotland Ltd.

Lines 42-43: "One of the main goals of community ecology is the identification and explanation of non-random patterns of species composition". Here, it would  be informative to cite some examples of this approach in community ecology, so that readers can find relevant case studies in the literature. For example, this recent example from waterbirds and aquatic plants: Lobato‑de Magalhães, T., et al. (2023). How on Earth did that get there? Natural and human vectors of aquatic macrophyte global distribution. Hydrobiologia, 850, 1515-1542.

Lines 43-45: Here, you should include a reference to beta-diversity and its use in conservation to support your statement. For example: Socolar, J.B., Gilroy, J.J., Kunin, W.E. & Edwards, D.P. (2016). How should beta-diversity inform biodiversity conservation? Trends in ecology & evolution, 31, 67-80.

Lines 45-46: You should provide a definition of beta-diversity here, to ensure that the reader understands the concept.

Line 57: To improve the language in this sentence, "Its" should be replaced with "This".

Lines 68-71: Range shifts, for example due to climate change, should also be mentioned here, as these are another mechanism through which the waterbird species pool at sites may change. Climate-induced range shifts have been documented for many waterbird species, for example: Nuijten, R.J.M., et al. (2020). Concurrent shifts in wintering distribution and phenology in migratory swans: individual and generational effects. Global Change Biology, 26, 4263-4275.

Line 90: You should start a new paragraph here.

Line 92: The word "is" should be deleted as it is not required in this sentence.

Line 98: To improve the language, in this sentence the word "avian" should be amended to "the avian community"

Line 99: In this sentence, "paid" should be amended to "received".

Line 99: It is still unclear what the "lake-island effect" means in this context. You should add a clear definition of this effect so that the reader can better understand the context and aims of your study.

Lines 99-104: I'm afraid I did not understand this sentence. It does not currently make sense in English. 

Lines 118-119: "The R software "iNEXT" package is used to evaluate the sample coverage test of the study area". This text is not very informative. How did you use iNEXT to evaluate sample coverage? What analysis did you carry out and what were the results (I could not link this text with any text in the results)? You should also include citations for both the R software (including the version number that you used) and the iNEXT package.

Line 120: The labels on the figure are small and difficult to read. A higher resolution image with larger labels should be included.

Lines 154-156: You should provide a reference for this classification of correlation coefficients.

Lines 165-166: You should include a list of the 58 species (along with their mean abundance at each sub-lake) as a supplementary information file.

Line 174: There seems to be an issue with the text here. Were some words deleted? It does not currently make sense.

Lines 180-182: "WNODF analysis showed that there is a significant nesting pattern distribution of waterbirds in Poyang Lake." This seems to contradict the results reported in the previous sentences, which reported that nesting was "lower than expected from the null model".

Line 184: To make the table legend more informative, you should include a definition of NODF, to help the reader interpret the values in the table.

Line 186: The Monte Carlo simulation should have also been mentioned in the methods. This should include the software used to run this analysis.

Lines 241-242: "These sub lakes were the most winter sites that harbor many species of birds including endangered birds". The words "the most" should be deleted. Can you give some examples of the endangered species that are supported here?

Line 242: The first letter of the new sentence should be capitalized.

Line 247-248: What is "blank habitat"?

Line 252: There is a typo here, "prsesnt" should be "present".

Comments on the Quality of English Language

There are major corrections needed for the language, including spelling and grammar, so that the meaning of the text can be understood. Unfortunately, in some sections of the manuscript I found it impossible to understand the message. I have listed examples of such corrections in my specific comments (in the previous section). I appreciate that you may not be writing in their first language, and so I hope that my suggested edits are helpful.

Author Response

Dear Reviewers I have answer your every question. i did point by point reply

Reviewer 3 Report

Comments and Suggestions for Authors

The study proposes to describe the beta diversity partitioning of wintering waterbird community in 24 sub-lakes of Poyang Lake. The topic is important to increase knowledge about the spatial distribution and behavior of these birds, and has the potential to support the conservation of these species. Unfortunately, there is a lack of clarity in the objectives of the study, in how these data were collected, in the analyzes conducted and in the presentation of the results. Furthermore, the authors do not discuss the results of the study, and there is a lack of biological interpretation of the analysis results.

The authors refer to the studied system as if it were several different communities. However, when examining the distances between habitat patches, it appears that the birds can easily move between patches. Thus, it is difficult to believe that they are different communities. The patches appear to be foraging and/or breeding sites, but all occupied by the same community. This description is not clear in the manuscript. This does not invalidate the analyzes carried out, but it would be more interesting to further detail the relationship between species richness and composition with the area and structure of the habitat patches or sub-lakes, focusing on the choice of patches by these birds. Unfortunately, the manuscript does not present possible ecological reasons for the patterns found, reducing the manuscript to a series of statistical results. A collection of numbers has little or no relevance to the knowledge of regional avifauna and ecology.

Some additional points that deserve attention follow below:

Introduction –

In the introduction, community measurement techniques are very detailed, but absolutely nothing is mentioned about birds. What biological hypotheses should be evaluated regarding the distribution of birds? What would the authors expect to find in a system like the one studied?

 Lines 30 - 34 – The study does not evaluate any anthropic impacts in the region, I question whether this sentence was necessary in the introduction.

Lines 37 - 39 – Is the area in which the study was conducted fragmented naturally or due to human influence? This is not clear even in the description of the study area. The reference cited at the end of the paragraph is about urbanization.

Lines 77 - 90 – Could the hypotheses presented in this part of the introduction even be applicable to the system studied? Could the extinction and colonization hypotheses be applied to this system? Distances between patches appear to be relatively short for waterfowl (at least for most). Couldn't individuals travel these distances with ease? In the case of this specific system, wouldn't these habitat patches just be preferred foraging sites?

Line 93 – what would be the “different factors that would be affecting the survival and spatial distribution of bird species?

Objectives – the presented questions are too general and vague. It would be necessary to better specificy the objectives of this study.

Methods and Data Analysis -  

Line 147-149 – The authors describe that “Spearman Correlation analysis was used to analyze the relationship between the bird species richness and the sub lake areas, and between the sub lake area and the habitat types richness”, but this is not in the objectives. In the conclusions, they present results of other correlations that then became justifications to corroborate the selective extinction hypothesis and habitat nestedness hypothesis.

 Results – The authors need to describe the bird species found in the different sub-lakes and present the biological meanings of the analysis results.

 Discussion – The discussion, as it stands, just repeats some results already presented in the Results item, as well as nd other results that were not previously presented in the Results item. The authors do not mention anything about birds (some mention of birds is made only in the conclusions, but very general and succinct). For example, the patches present four different habitat types and the authors showed a (weak) correlation between bird richness and the number of bird types. The variation in habitat richness between sub lakes is very small (only 3 or 4 habitat types), which makes the analysis very questionable. But even so, are there species that have been observed in sub lakes with four habitat types that have not been observed in sub lakes with three types? If so, which would they be and why?

Suggestions – The suggestion is not directly related to the study, and is only subjective. It should be rewritten or deleted.

 Conclusions – The conclusion is also very vague and presents repetitions of results or extra results, which were not in the Results item.

Author Response

Dear Reviewer I have done point by point reply please have a look

Round 2

Reviewer 3 Report

Comments and Suggestions for Authors

The manuscript has improved in this second version. The study is more concise, direct, and clearer when compared to the first version. However, some small adjustments are still necessary, especially in the Abstract. In this item, the authors should suppress details of statistical analyses and focus on the main biological results, and the importance of the lake for bird conservation. Despite these small errors, the study presents very interesting results about the relationship between bird communities and the characteristics of habitats used for foraging.

Still in the abstract, the authors mention that the birds were observed for five consecutive winter seasons, but in the table caption in the supplementary material it says four consecutive Years. What happened to the data from the fifth year?

​Throughout the text, several small typographical and spelling erros, as well as some grammar errors are still found, that make it difficult for the reader to understand. For example:

Page 1, line 18 – consetive instead consecutive

Page 1 Line 20-21 – The word Family is missing in this sentence “58 wintering water bird species belonging to 9 orders and 15 ???? were observed from 2016 to 2020

The supplementary material should include the average number of observed individuals of each species instead of the total, since a single individual may have been observed in different years.

Introduction

Page 1 – line 36, reference 12 is cited incorrectly. This reference is about beta diversity and not specifically about waterbirds.

Page 2 – line 49, 50, This sentence is quite confusing.

Page 2 – line 74,75, This sentence is completely out of context and should be deleted.

Results

Page 6 – line 170 – the citation to Table 2 should come after “lake area”.

Page 8 – line 194 – it is not necessary to repeat results of statistical analyses in the discussion.

Comments on the Quality of English Language

Throughout the text, several small typographical and spelling erros, as well as some grammar errors are still found, that make it difficult for the reader to understand.

Author Response

dear Reviewer I did all the changes
